# Enriching Egg Quality of Laying Hens from the Canary Islands by Feeding with *Echium* Oil

**DOI:** 10.3390/foods15010018

**Published:** 2025-12-21

**Authors:** Jesús Villora, Alexandr Torres, María Fresno, Sergio Álvarez, Nieves Guadalupe Acosta, José Antonio Pérez, Covadonga Rodríguez

**Affiliations:** 1Departamento de Biología Animal, Edafología y Geología, Universidad de La Laguna, Avenida Astrofísico Francisco Sánchez s/n, 38206 San Cristóbal de La Laguna, Tenerife, Spain; ngacosta@gmail.com (N.G.A.); janperez@ull.edu.es (J.A.P.); covarodr@ull.edu.es (C.R.); 2Unidad de Producción Animal, Pastos y Forrajes en Zonas Áridas y Subtropicales, Instituto Canario de Investigaciones Agrarias, 38200 San Cristóbal de La Laguna, Tenerife, Spain; atorresk@icia.es (A.T.); mfresno@icia.es (M.F.); salvarez@icia.es (S.Á.)

**Keywords:** dual-purpose hens, egg quality, omega-3 PUFA, *Echium* oil, stearidonic acid

## Abstract

*Echium* species, abundant in the Canary Islands, contain unique fatty acids (FA) such as stearidonic acid (SDA; 18:4n-3) and γ-linolenic acid (GLA; 18:3n-6), which may improve egg quality while valorizing local genetic resources. This study evaluated the effects of *Echium plantaegineum* oil (EO) compared with linseed oil (LO) and soybean oil (SO) on productive performance, egg quality, sensory traits, and yolk fatty acid profile. Forty-eight hens from the Canary Islands were fed for 31 days with diets supplemented with 1.25% SO (SO-d), 1.1% LO + 0.15% beef tallow (LO-d), and 1% EO + 0.25% LO (EO-d). LO supplementation reduced laying rate and egg mass with respect to SO, increasing feed conversion ratio (FCR), whereas EO produced slightly lighter eggs compared to the SO group but with normal yolk proportion and shell traits. EO markedly increased egg yolk deposition of SDA, eicosapentaenoic acid (EPA; 20:5n-3), docosapentaenoic acid (n-3 DPA; 22:5n-3), and docosahexaenoic acid (DHA; 22:6n-3), while lowering the n-6/n-3 ratio and thrombogenic index (TI). No differences were observed in the evaluated sensory attributes among treatments. In conclusion, dietary inclusion of EO effectively enriches eggs with n-3 LC-PUFA without negatively affecting sensory quality, supporting its potential use as a functional ingredient in laying hen diets.

## 1. Introduction

Eggs are widely recognized as a highly nutritious food, providing not only high-quality proteins but also an array of essential vitamins, minerals, and bioactive compounds with significant physiological functions [1]. However, despite their overall nutritional value, the fatty acid (FA) profile of conventional eggs is generally considered suboptimal from a human health perspective. Commercial eggs typically exhibit a high content of omega-6 polyunsaturated fatty acids (n-6 PUFA) and a relatively low proportion of omega-3 (n-3) PUFA, contributing to an unbalanced n-6/n-3 ratio [2]. Dietary n-6/n-3 imbalance has been linked to the development of various chronic pathologies, including cardiovascular diseases, metabolic syndrome, inflammatory disorders, and neurodegenerative conditions [3,4].

N-3 long-chain polyunsaturated fatty acids (n-3 LC-PUFA), particularly eicosapentaenoic acid (EPA; 20:5n-3), docosapentaenoic acid (n-3 DPA; 22:5n-3), and docosahexaenoic acid (DHA; 22:6n-3), are well documented for their crucial roles in human health. These FAs are fundamental for cardiovascular protection, optimal brain function, neurodevelopment, immune modulation, and the reduction of systemic inflammation [5,6]. However, their presence in the Western diet remains significantly below recommended levels, primarily because the main dietary sources, including oily fish and marine products, are not universally accessible, affordable, or environmentally sustainable [7].

Consequently, the search for enriched land-based sources of n-3 LC-PUFA has intensified in recent years. In this context, poultry products, including eggs, offer a highly promising avenue. Eggs are affordable, widely consumed globally, and culturally accepted across populations. Furthermore, chickens have demonstrated a remarkable capacity to metabolize 18C n-3 precursors, such as α-linolenic acid (ALA; 18:3n-3), into bioactive n-3 LC-PUFA like EPA, DPA, and DHA [8,9]. This metabolic efficiency positions eggs as a viable, sustainable, and scalable vehicle for delivering n-3 LC-PUFA to the human diet, with less environmental impact compared to marine-based sources.

Numerous studies have demonstrated that the FA composition of eggs can be substantially modified through dietary intervention, mainly by incorporating marine sources containing preformed LC-PUFA [10,11] or vegetables rich in ALA [12,13]. The use of fish oil in animal feed represents a significant economic and environmental cost, being one of the most expensive components of livestock diets. Moreover, fish oil is a valuable component of the human diet, and its direct consumption by humans is metabolically more efficient than its indirect use through animal products [14]. The biosynthesis of EPA and DHA from ALA requires a sequence of elongation and desaturation reactions. The process begins with the Δ6 desaturase enzyme (FADS2) converting ALA into stearidonic acid (SDA; 18:4n-3). This first desaturation step is widely recognized as the main rate-limiting point in the pathway and is further constrained by the competition for the same enzyme with linoleic acid (LA; 18:2n-6) to form γ-linolenic acid (GLA; 18:3n-6) [8], or even with 24:5n-3 to form 24:6n-3 before the last step of peroxisomal beta oxidation to finally lead to DHA [15].

Thus, incorporating vegetable sources rich in SDA could be a more efficient approach to increase the deposition of n-3 LC-PUFA in poultry products [16]. In this context, several studies have explored the impact of SDA supplementation on the lipid profile of egg yolk. For instance, Elkin et al. [17] demonstrated that the inclusion of SDA-enriched soybean oil (SO) in the diet of Line W-36 hens was more effective than linseed oil (LO) in enhancing n-3 PUFA deposition. However, the achieved levels of n-3 LC-PUFA did not reach those obtained with direct supplementation with fish oil, which provides preformed EPA and DHA. Similarly, El-Zenary et al. [18] reported that dietary supplementation with Ahiflower^®^ oil, a plant source naturally rich in SDA, was more effective than LO in increasing the deposition of n-3 LC-PUFA, such as eicosatetraenoic acid (ETA; 20:4n-3), EPA, DPA, and DHA, in the egg yolk of Hy-Line W-36 White Leghorn hens.

Among plant-derived oils, species from the *Echium* genus (Boraginaceae) are of particular interest due to their unique FA profile. *Echium plantagineum* oil (EO), in particular, contains high concentrations of SDA [19,20], potentially bypassing the afore-mentioned rate-limiting Δ6 desaturation steps [21]. Additionally, EO contains γ-linolenic acid (GLA; 18:3n-6), which can be elongated to dihomo-γ-linolenic acid (DGLA; 20:3n-6) and metabolized via cyclooxygenase and lipoxygenase to generate metabolites that suppress inflammation and decrease stress symptoms [22].

The Canary Islands are considered a global hotspot of biodiversity, with an exceptional richness in *Echium* species, including 28 endemic taxa [23]. Moreover, the native chicken from the Canaries is a well-adapted local genetic resource, ideal for low-input, extensive systems thanks to its dual-purpose capacity for both egg and meat production [24]. This offers a valuable opportunity to explore the use of animal resources and feed ingredients for functional and local food production.

Recent research by Villora et al. [25] demonstrated the superior effectiveness of EO compared to LO in enhancing the deposition of n-3 LC-PUFA, particularly n-3 DPA and DHA, in the muscle tissue of native Canarian chickens. However, to date, no studies have investigated the impact of EO supplementation on eggs. Therefore, the aim of this study was to evaluate the effect of dietary supplementation with EO, in comparison with LO and the widely commercially used SO, on productive performance, egg physicochemical quality, yolk lipid and FA profiles, and sensory quality of eggs from native laying hens from the Canary Islands.

## 2. Materials and Methods

### 2.1. Ethics Statements

All animal procedures were conducted in accordance with the European Directive 2010/63/EU and the Spanish RD53/2013 on the protection of animals used for scientific purposes. In this study, the animals were reared following standard commercial farm practices without applying any experimental or invasive procedure. The animal study protocol was approved by the Institutional Animal Welfare and Ethics Review Committee of Instituto Canario de Investigaciones Agrarias/Canary Islands Government (approval code: CEEA-ICIA-2025-001, date: 7 March 2025).

### 2.2. Animals and Diets

The trial was conducted at the experimental farm of the Instituto Canario de Investigaciones Agrarias in Tenerife, Spain. A total of 48 hens from the Canary Islands (Figure 1), aged 52 weeks, were allocated into three balanced groups (*n* = 16 birds per group) based on their body weight.

Birds were housed in separate pens, each with access to an outdoor area of approximately 25 m^2^. This space was fully enclosed, delimited, and covered exclusively with sand, without any vegetation, insects, or other potential feed sources. Each group was fed a different diet, prepared by applying one of three distinct vegetable oil blends onto a common low-fat cereal base (Grupo Capisa, Tenerife, Spain). The FA profile of each added fat source is represented in Table 1.

This resulted in the formation of three experimental dietary groups: soy oil diet (SO-d), supplemented with 1.25% soybean oil (SO), which served as the control group; linseed oil diet (LO-d), supplemented with 1.1% linseed oil (LO) + 0.15% beef tallow; and *Echium* oil diet (EO-d), containing 1% *Echium plantagineum* oil + 0.25% LO, as detailed in Table 2. LO-d and EO-d were formulated to have similar n-6/n-3 ratios (2.2 and 2.9, respectively), while SO-d presented a significantly higher n-6/n-3 ratio of 13.3, resembling that of commercial hen pellets. All diets were isoenergetic, isoproteic, and isolipidic, and were provided ad libitum over a period of 31 days. LO was combined with beef tallow, and EO with LO, to equilibrate the n-6/n-3 ratio and energy content across diets. This formulation ensured comparable lipid profiles between treatments, allowing biological responses to be attributed specifically to the SDA provided by EO rather than to differences in n-6/n-3 ratios.

To minimize potential confounding by pen-specific environmental factors, the birds receiving each dietary treatment were rotated among the three available pens at 10 day intervals. All birds within a dietary group were moved together and maintained on the same diet throughout the study. As a result, each dietary treatment occupied each pen for an equal amount of time over the course of the experiment. All pens were managed under identical environmental and husbandry conditions, and the order of pen rotation was the same for the three dietary groups. During the trial, feed samples were collected in triplicate to analyze their lipid and FA composition (Table 2).

All hens remained healthy throughout the trial, and no mortality or adverse events occurred during the feeding period. Throughout the trial, hens were handled following standard welfare practices to minimize stress, pain, or discomfort. Potential confounding, pen-specific factors were minimized.

### 2.3. Productive Performance

All hens were individually weighed at the beginning and at the end of the dietary trial to assess body weight changes over the experimental period using a portable poultry scale BIT PS 3.0 (Broring Technology GmbH, Oldenburg, Germany). Feed intake was recorded weekly for each experimental group by measuring the amount of feed offered and the residual feed. Data are expressed as average group feed intake (g) per week. Eggs were collected daily from each group, and the total number of eggs was recorded to calculate laying rate (%), which is defined as the percentage of eggs laid per hen per day, reflecting the productivity of the laying hens over time. The total amount of eggs was weighed using a digital scale Kern PCB 1000-2 (Kern and Sohn, Balingen, Germany). The total weight was divided by the total number of eggs to calculate egg mass. The weekly feed conversion ratio (FCR) was calculated as the total feed consumed divided by the total egg mass produced, and expressed as kg of feed/kg of eggs. Egg mass was determined by multiplying the laying percentage by the average weight of eggs (g) and divided by 100, representing the average amount of egg material produced per hen per day.

### 2.4. Egg Quality

Eggs for physical and chemical analyses were collected on days 28–31 of the feeding trial (*n* = 16 per dietary treatment). The morphological structure of the egg components was determined by weighing the yolk, eggshell, and egg white individually and expressed as a percentage of the whole egg.

The color parameters of raw egg yolk and eggshell were measured using a CR-400 Chroma Meter (Minolta Camera Co. Ltd., Osaka, Japan) in the CIELAB space: lightness (L*); redness (a*); and yellowness (b*). For each sample, three measurements were taken at the same position. The colorimeter was calibrated with a standard white tile.

Egg white and yolk pH were determined in triplicate using a penetration pH electrode by means of a pH meter GLP 21 (Crison Instruments SA, Barcelona, Spain).

Eggshell thickness was assessed three times on the longest and flattest region of each egg with a stainless-steel caliper (accuracy 0.01 mm) and the mean value was recorded. Measurements were consistently taken on the same location of the egg, specifically on the longest face at the equatorial region.

### 2.5. Lipid Composition

Eggs for lipid analyses were collected on days 28–31 of the feeding trial. The total lipid (TL) content of diets (*n* = 3) and egg yolks (*n* = 5) was extracted according to [26] with small modifications described by [27].

The lipid classes (LC) profile from egg yolks (*n* = 5) was determined from 0.02 mg of TL extracts. LC were analyzed by high-performance thin-layer chromatography (HPTLC) in a one-dimensional double-development system to separate polar and neutral lipids [28]. Lipid classes were identified by comparison to external lipid standards placed on the same HPTLC plate and quantified by calibrated densitometry using a dual wavelength flying spot scanner CAMAG TLC Visualizer (Camag, Muttenz, Switzerland), as described by [27].

Fatty acid methyl esters (FAMEs) from diets (*n* = 3) and egg yolks (*n* = 5) were obtained by acid-catalyzed transmethylation of 1 mg TL aliquots. FAMEs were purified by thin-layer chromatography [29] using pre-coated TLC plates SIL G-25 (20 cm × 20 cm; Macherey-Nagel GmbH & Co. KG, Düren, Germany) and resolved with hexane/diethyl ether/acetic acid (90:10:1, *v*/*v*) as detailed by Galindo et al. [30]. For absolute quantification of FAs (mg FA per 100 g), 50 µg of nonadecanoic acid (19:0) was added to the lipid extracts prior to transmethylation. Purified FAMEs were analyzed using a TRACE-GC Ultra gas chromatograph (Thermo Fisher Scientific Inc., Waltham, MA, USA) equipped with an on-column injector, a flame ionization detector and a fused silica capillary column (Supelcowax TM 10; Sigma-Aldrich Co., St. Louis, MO, USA). FAMEs were identified by comparison of their retention times with that of a commercial standard mixture (FAME Mix C4-C24 and PUFA N° 3 from menhaden oil; Supelco Inc., Bellefonte, PA, USA) and a well-characterized cod roe FAME sample. When necessary, the unequivocal identity of a FA was also assessed by GC-MS. Results are expressed as the percentage of total FAs for the diets, and as mg FA per 100 g of fresh yolk.

### 2.6. Nutritional Indices

Nutritional quality of eggs was assessed for all dietary treatments in terms of egg yolk FA composition, calculating the atherogenic (AI) and thrombogenic indices (TI) [31], as well as the ratio between hypocholesterolemic and hypercholesterolemic FAs (hH) [32], as follows:AI = [12:0 + (4 × 14:0) + 16:0]/(∑MUFAs + ∑n-6 PUFAs + ∑n-3 PUFAs)TI = (14:0 + 16:0 + 18:0)/(0.5 × ∑MUFAs + 0.5 × ∑n-6 PUFAs + 3 × ∑n-3 PUFAs + n-3/n-6 ratio)hH = (18:1n-9 + 18:2n-6 + 20:4n-6 + 18:3n-3 + 20:5n-3 + 22:5n-3 + 22:6n-3)/(14:0 + 16:0)

### 2.7. Sensory Analysis

Eggs used for the sensory evaluation were collected from the three experimental groups exclusively between days 28 and 31 of the feeding trial. After collecting, the eggs were individually identified and stored under refrigeration (4 °C) for three days prior to analysis. All eggs were hard-boiled following a standardized procedure: once the water reached boiling point, the eggs were carefully placed into the pot and cooked for 8 min. Upon completion, the eggs were immediately cooled in an ice-water bath to halt further cooking. To ensure consistency in doneness, eggs from all three dietary groups were boiled simultaneously. After cooling, the eggs were peeled, and each unit was longitudinally cut in half.

The sensory evaluation was conducted by a trained panel consisting of 10 panelists from the Instituto Canario de Investigaciones Agrarias with experience in the evaluation of egg profiles, following the instructions given by the norm [33]. Each panelist received a plate containing one half of an egg from each dietary treatment. The samples were coded with random three-digit numbers to ensure a blind evaluation.

Panelists assessed sensory attributes related to intensity (egg odor, off-odor, egg flavor, off-flavor, aftertaste, off-aftertaste, yolk color) and acceptance (odor, flavor, aftertaste, yolk color, texture, overall acceptability). Intensity attributes were rated using a 9-point intensity scale, while acceptance attributes were scored using a 9-point hedonic scale. Prior to the evaluation, all panelists provided written informed consent to participate in the study.

### 2.8. Statistical Analysis

All the dependent variables studied were examined for normal distribution by the Shapiro–Wilk test and for homogeneity of the variances with the Levene test prior to analysis. When necessary, variance-stabilizing transformations (arcsine and logarithm) were applied. Subsequently, if both assumptions were satisfied, a one-way ANOVA followed by Tukey’s post hoc test was used to assess the effect of the diet (SO-d, LO-d, and EO-d). Welch test followed by the Dunnett T3 test was performed for non-homoscedastic data, and Kruskal–Wallis non-parametric test was applied in case of non-normal distribution, followed by the pair-wise comparisons Mann–Whitney test with Bonferroni correction. PUFAs were additionally submitted to Principal Component Analysis (PCA), and factor scores were subsequently analyzed by ANOVA. Statistical significance was set at *p* < 0.05. All statistical analyses were carried out using IBM SPSS statistics 25.0 for Windows (SPSS Inc., Armonk, NY, USA).

## 3. Results

### 3.1. Feed Intake, Efficiency and Egg Production Traits

At the end of the experimental period, there were no significant differences in hens’ body weight and feed intake among the dietary groups (Table 3). However, dietary LO negatively affected productive parameters, as evidenced by a lower laying rate and reduced egg mass, which in turn led to a higher feed conversion ratio (FCR) when compared to the SO-d.

### 3.2. Egg Physical and pH Characterization

As displayed in Table 4, egg weight was significantly affected by dietary treatment. Feeding EO resulted in lighter eggs compared to SO-d. Nevertheless, hens fed the SO-d produced eggs with a lower yolk proportion than those fed the LO-d (*p* < 0.05). The yolk from EO-hens exhibited greater redness than that from LO-hens, whereas eggs from the LO-d birds showed the lowest shell thickness. By contrast, no significant differences (*p* > 0.05) were observed for shape index, egg white and eggshell proportions, pH values, and eggshell color.

### 3.3. Lipid Content and Lipid Class Composition of Egg Yolk

The total lipid (TL) content of egg yolks was not significantly affected by the dietary treatments, averaging around 31% of fresh weight across all groups (Table 5). The inclusion of EO reduced total polar lipids (TPL) while increasing total neutral lipids (TNL) (*p* < 0.05). Major phospholipids showed diet-dependent differences. Phosphatidylcholine (PC) was significantly lower in eggs from hens fed the EO-d compared to both the SO-d and LO-d. Eggs from the EO-d group presented significantly lower phosphatidylethanolamine (PE) levels than those fed LO-d and higher diacylglycerols (DAG) and sterol esters (SE) compared to SO-d.

### 3.4. Main Fatty Acid Composition of Egg Yolk and Nutritional Indices

The proportions of SFA and MUFA in egg yolk were not significantly affected by the dietary treatment (Table 6). Although total n-6 PUFA levels, including LA and ARA, remained unchanged, both LO and EO diets markedly reduced the content of 22:5n-6 (n-6 DPA) to less than half that of SO-d. The inclusion of EO notably increased the deposition of n-3 PUFAs in the yolk, particularly SDA, and n-3 LC-PUFAs such as 20:3n-3, EPA, DPA, and DHA. Additionally, the EO diet enhanced the levels of GLA and DGLA more efficiently than the LO diet. Notably, EO reduced TI with respect to SO-fed hens.

The PCA for PUFAs revealed that PC1 and PC2 accounted for 53.46 and 17.49% of the variance, respectively (Figure 2). PC1 was highly correlated with the n-3 LC-PUFA—EPA (0.98), n-3 DPA (0.95), and DHA (0.93)—and clearly separated the data into three distinct clusters.

### 3.5. Egg Sensory Analysis

The sensory attributes of eggs from hens fed diets enriched with SO, LO, or EO are presented in Table 7. No significant differences (*p* > 0.05) were observed among treatments for any of the sensory parameters evaluated, either in terms of intensity or acceptance.

## 4. Discussion

Given the growing interest in functional foods and the enrichment of animal-derived products with health-promoting FA, the present study was designed to evaluate the effects of supplementing the diets of a local hen breed from the Canary Islands with different plant-based oils, including a native oil source rich in SDA. Supplementation with EO showed a strong capacity to enrich egg yolk with n-3 LC-PUFA, including EPA and DHA. Although some variation in laying performance was observed, especially in the LO-d group, the overall results highlight the potential of EO as an effective alternative for the nutritional enhancement of eggs from local hen breeds.

After 30 days of feeding, the final body weight of hens remained stable among treatments (around 2.4 kg, Table 3). These values are consistent with previous data reported for the native hen population of the Canary Islands [34]. In our study, LO-d hens showed a lower laying rate compared to those fed SO-d (Table 3). In agreement with our findings, Petrovic et al. [35] also reported a reduction in laying rate when supplementing 2% linseed oil in Lohmann Brown hybrid hens. Conversely, other authors did not observe any significant variations in laying performance when using flaxseed-rich diets [12,36]. Egg mass was also affected by the dietary treatment. The lowest value was registered in the LO-d group (15.98 g), and consequently, also the highest FCR (10.57). El-Zenary et al. [18] similarly found a reduction in egg mass in Hy-Line W-36 White Leghorn hens fed 0.75% LO or Ahiflower compared to control hens, although the number of eggs laid and hen-day egg production remained unaffected.

In a recent review, Irawan et al. [37] noted that increased ALA intake tends to reduce egg production and to increase FCR. Moreover, it has been summarized that the dietary inclusion of 2–3% LO generally does not impair productive performance, but higher levels (≥5%) may significantly decrease body weight and laying rate. This effect has been attributed to the high efficiency of n-3 PUFA deposition and the potential presence of antinutritional factors in LO [38]. In our study, the amount of LO included was relatively low (1.10%). However, under our experimental conditions, the greater susceptibility of LO-d to oxidation, due to its higher number of double bonds compared to SO-d (i.e., 415.2% vs. 82.0% total n-3 PUFA, respectively; see Table 2 for details), cannot be completely ruled out as a contributing factor to the adverse effects on laying performance observed with this dietary treatment. In a previous study, similar inclusion levels of LO and EO neither affected the growth performance of Canarian cockerels [25]. These discrepancies with literature could be partially explained by the broodiness behavior typical of Canarian hens. This behavior is common among indigenous breeds but has been largely eliminated in commercial laying strains developed by the modern poultry industry [39]. Broodiness has been associated with reduced egg production, as hens typically cease laying for approximately 20 days during the brooding period [40].

Under our experimental conditions, hens fed SO-d laid heavier eggs compared to those fed the EO diet (Table 4). However, the highest yolk percentage was observed in the LO-d group (34.41%). In general, the yolk proportion in eggs from Canary Islands hens was higher than that reported for other dual-purpose genotypes [41,42], and similar to values obtained for the Canarian genotype in a previous study [43]. Regarding pH values of yolk and egg white, the absence of significant dietary effects is consistent with the findings of Batkowska et al. [44], who reported that supplementation with SO and LO did not influence yolk or egg white pH in fresh eggs. The pH values of egg fractions in the present work fall well within the normal values described in the literature. Thus, the optimal pH of egg white ranges from 7.5 to 8.5, increasing during storage to values up to 9.5 [45], while the pH of freshly laid yolk is approximately 6.0 [46]. Overall, the yolks presented very high a* values, particularly in the EO birds (12.50, Table 4), exceeding those recently reported in eggs from the Canarian genotype fed with a commercial layer feed by Sigut et al. [43]. Some studies have reported no effect on yolk color following the inclusion of 13.5% full-fat linseed oil in the diet [47], whereas others have shown that dietary inclusion of linseed and fish oil can influence yolk pigmentation [48,49]. In addition, our findings are consistent with those reported by Kralik et al. [50], who observed a reduction in shell thickness following dietary inclusion of 2% linseed oil combined with 1.5% rapeseed oil and 1.5% fish oil. The present results revealed a considerable inter-individual variability in egg production and quality, which may be due to genetic heterogeneity and lack of intensive selection for uniform production traits, as observed in other local breeds. Such variability should be considered when evaluating performance and interpreting data from native poultry populations.

Despite the absence of variation in total yolk lipid content (~31%, Table 5), EO feeding resulted in a certain shift in the lipid class distribution, increasing the proportion of neutral lipids (78.66%) while reducing that of polar lipids (21.34%). This distribution is very similar to that reported in yolks from ISA Brown hens supplemented with the microalga *Nannochloropsis gaditana* [51]. In particular, EO decreased the relative abundance of PC and PE phospholipids while enhancing DAG and SE levels. A comparable trend was observed in the breast meat of Canarian cockerels fed 2% EO [25]. Corrales-Retana et al. [2] demonstrated that DHA tends to accumulate preferentially in neutral lipids, perhaps as an attempt to make it more accessible to the embryo. In line with this, EO-d deposited higher amounts of DHA into egg yolk compared to the other treatments (Table 6), as will be further discussed. Moreover, the observed increase in SE could be associated with cholesterol esterification processes, due to a modulation of cholesterol metabolism and transport, and changes in gene expression involving PPARα and SREBP-2, as previously described [52].

The inclusion of LO and EO did not reduce the levels of n-6 PUFA, including LA and ARA, compared to SO (Table 6). In contrast, previous studies have reported reductions in LA and ARA in egg yolk when using SDA-enriched soybean oil [17] or Ahiflower oil [18]. These discrepancies are likely related to the high amounts of LA present in our experimental diets (850–1000 mg/100 g). Nevertheless, both LO and EO effectively decreased n-6 DPA deposition, a final desaturation and elongation product of ARA. The production of this FA metabolically competes for the synthesis of n-3 DPA, which contributes to anti-inflammatory, cardiovascular, and cognitive benefits [53]. EO enhanced the deposition of GLA and its elongation product DGLA, consistent with findings in breast meat of Canarian cockerels [25]. GLA was present in the EO and is formed also from LA via ∆6-desaturase and elongated to DGLA, but only a small fraction of DGLA is converted into ARA due to the limited activity of ∆5-desaturase [54]. As a result, DGLA accumulates after dietary GLA supplementation [22]. DGLA is further metabolized through COX and LOX pathways into prostaglandins with anti-inflammatory, vasodilatory, blood pressure–lowering, antitumor, and anti-proliferative effects [55].

Although the EO-d diet supplied the highest amount of SDA (76.78 mg/100 g vs. ~2 mg/100 g in LO-d and SO-d), its accumulation in egg yolk was proportionally much lower (66.81 mg/100 g vs. 36.84 and 15.13 mg/100 g, respectively). Elkin et al. [17] reported that, unlike ALA, which tends to be more evenly distributed, most of the consumed SDA is likely oxidized and/or preferentially transported to somatic tissues such as adipose. These authors suggested that in chickens, an alternative pathway for SDA transport may exist, involving its secretion from the liver either as an albumin-bound non-esterified FA or as part of a specific subclass of VLDL particles containing apolipoprotein VLDL-IV (Apo-IV). In the present study, the FA profile of extra-ovarian tissues was not assessed, and therefore, these mechanisms cannot be confirmed. An alternative and feasible hypothesis is that most of the consumed SDA was more efficiently converted into n-3 LC-PUFA than ALA, and subsequently transported to the yolk, as reflected in its FA composition, which is richer in EPA, n-3 DPA, and DHA.

To date, few experiments have evaluated the dietary supplementation of SDA for the enrichment of hen eggs. Elkin et al. [17] first demonstrated the superior efficacy of combining ALA + SDA, compared to ALA alone, in enriching eggs with n-3 LC-PUFA. However, in the same study they found that supplementation with SDA alone resulted in only 39% of the n-3 PUFA deposition achieved when hens were supplemented with an equivalent amount of fish oil. More recently, El-Zenary et al. [18] concluded that SDA increased n-3 LC-PUFA more effectively than ALA when treatments were equivalent in total n-3 FA, although this effect diminished for DHA. Interestingly, in the present study, EO supplementation most effectively enriched egg yolk with n-3 PUFA, including 20:3n-3, EPA, n-3 DPA, and DHA (Table 6). Notably, DHA was the predominant n-3 PUFA in yolks from EO birds, in contrast to chicken meat, where DPA accumulated to a greater extent than DHA [25]. Our results are supported by a recent review indicating that laying hens are able to convert DPA to DHA with relatively high efficiency [56].

Considering that the average egg weight from Canarian EO fed hens was ~60 g, with yolk accounting for 32.36% of the egg (Table 4), each yolk from the EO-d group would provide approximately 81 mg of DHA and 87 mg of EPA + DHA. Given the recommended daily intake of 500 mg of EPA + DHA for cardiovascular health [57], the consumption of EO eggs could represent a valuable contribution to meeting this requirement. Moreover, according to current EU regulations on nutrition and health claims [58], eggs can be labeled as a “source of omega-3 EPA + DHA” if they contain at least 40 mg per 100 g and per 100 kcal. Based on our findings, eggs from Canarian hens fed EO-d clearly meet this criterion.

Another important parameter frequently used to evaluate the nutritional quality of food is the n-6/n-3 ratio, which should ideally approach 1:1 [59]. In our study, LO- and EO-supplemented diets richer in n-3 PUFA reduced this ratio from 10.88 in SO-d eggs to 4.75 and 3.12, respectively (Table 6). It has been proposed that the optimal n-6/n-3 ratio should range between 1:1 and 5:1 to maintain a healthy balance [60]. In this regard, LO- and EO-eggs contributed to achieve this goal. Additionally, the thrombogenic index (TI) was used to further characterize the nutritional quality of egg FA composition on cardiovascular health, as it reflects the balance between pro-thrombogenic and anti-thrombogenic effects [61]. In our trial, the TI value was the lowest in eggs from EO-fed hens (0.79 vs. 0.82 and 0.93 for LO- and SO-fed hens, respectively) (Table 6). This value was lower than those reported for hens under conventional diets [62] and with olive pulp supplementation [63], but higher than those obtained with a combination of LO and fish oils [1]. Lower TI values have been consistently associated with positive health benefits, including reduced risk of atherosclerosis and atrial fibrillation [64,65].

A review of the existing literature revealed no studies assessing the sensory quality of eggs produced by hens fed diets rich in SDA. Nonetheless, several studies have reported adverse effects on egg sensory quality when using dietary sources rich in n-3 PUFA [66,67,68]. According to the trained panel, the sensory characteristics showed no significant differences between eggs from hens fed n-3 PUFA–rich oils and those fed diets with higher proportions of n-6 FA. The odor and flavor values obtained with the SO-d reatment were very similar to those reported by Sigut et al. [43] for eggs from Canary hens. Although the LO and EO treatments resulted in slightly lower evaluations of egg odor, flavor, and overall acceptability, these differences were not statistically significant, indicating no rejection by the panelists. Consistently, Feng et al. [69] reported that the acceptance threshold for DHA in eggs from hens supplemented with fish oil was 4.27 mg/g. Since the eggs from hens fed the EO-d diet in our study contained 4.16 mg/g of DHA, this value falls below the established threshold, which supports the absence of rejection by the panelists.

## 5. Conclusions

Dietary inclusion of EO did not affect hen body weight and resulted in slightly lighter eggs, while yolk proportion and shell traits remained normal. Moreover, EO produced healthier eggs due to its higher content of certain n-6 PUFA with anti-inflammatory properties, such as GLA and DGLA. Likewise, it also enhanced the eggs’ levels of n-3 PUFAs, including 20:3n-3, EPA, n-3 DPA, and DHA and reduced n-6/n-3 ratio and TI. Concurrently, supplementing hens’ diet with 1% EO did not affect the sensory quality of eggs. In light of the growing consumer demand for healthier and more sustainable animal products, enriching eggs with n-3 LC-PUFAs from plant-based sources such as EO offers a promising strategy to reduce reliance on marine resources. Furthermore, this study highlights the combined use of a vegetable oil derived from an endemic plant and local dual-purpose, slow-growing breeds, such as native hens from the Canary Islands. This integrated approach contributes to preserving local genetic resources, enhances the nutritional quality of animal products, and reduces reliance on external inputs in the Canary Islands.

## Figures and Tables

**Figure 1 foods-15-00018-f001:**
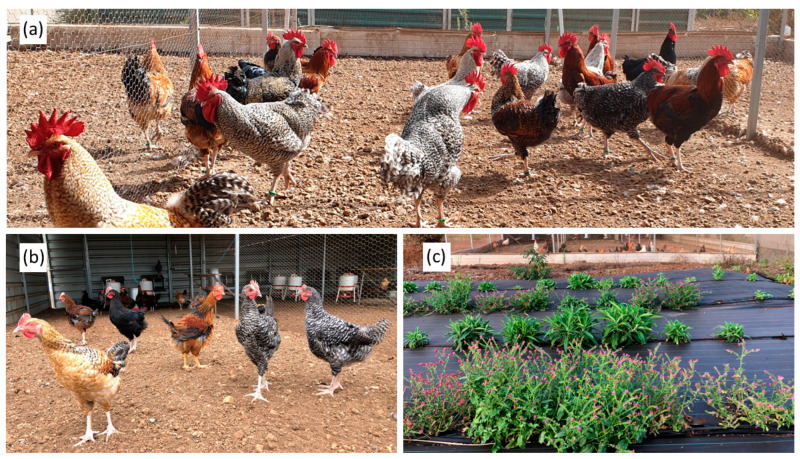
Male (**a**) and female (**b**) specimens of the Canarian chicken genotype, and *Echium plantagineum* plants (**c**).

**Figure 2 foods-15-00018-f002:**
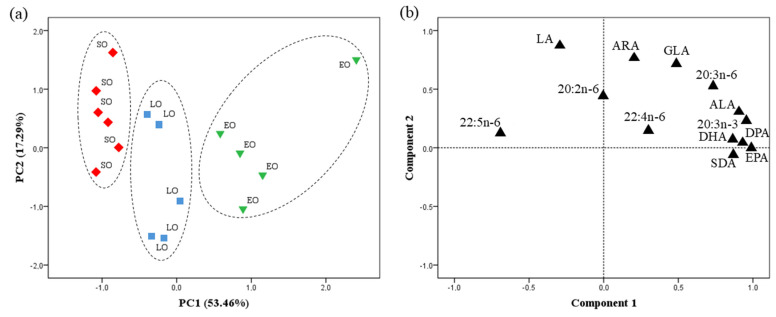
Principal component analysis (PCA) of PUFAs from egg yolk of laying hens from the Canary Islands. (**a**) Factor score plot for PC1 and PC2. [(◆) SO-d = soybean oil diet; (■) LO-d = linseed oil diet; (▼) EO-d = *Echium* oil diet]. Dashed line (---) represents different clusters for PC1 (*p* < 0.05). (**b**) Component loading plot for the PCA illustrates the correlation between each individual PUFA and the principal components PC1 and PC2. LA, 18:2n-6; GLA, 18:3n-6; ARA, 20:4n-6; ALA, 18:3n-3; SDA, 18:4n-3; EPA, 20:5n-3; DPA, 22:5n-3; DHA, 22:6n-3.

**Table 1 foods-15-00018-t001:** Main fatty acid composition (% of total fatty acids) of the fat sources used to supplement the experimental diets.

Item	Beef Tallow	SO	LO	EO
∑SFA	36.18	15.78	10.27	11.46
16:0	22.23	10.62	5.87	7.25
18:0	11.76	4.01	3.71	3.79
∑MUFA	48.53	23.28	19.97	17.48
18:1n-9	39.97	21.20	17.80	14.47
18:1n-7	3.72	1.54	1.40	1.08
∑n-6 PUFA	12.35	53.31	16.05	27.47
18:2n-6	11.32	53.31	15.99	16.32
18:3n-6	0.07	nd	nd	11.08
∑n-3 PUFA	0.85	6.59	48.31	42.12
18:3n-3	0.72	6.47	48.08	29.32
18:4n-3	nd	0.12	0.23	12.78
n-6/n-3	14.55	8.09	0.33	0.65

SFA, saturated fatty acids; MUFA, monounsaturated fatty acids; PUFA, polyunsaturated fatty acids; nd, not detected. Totals include other minor components not shown.

**Table 2 foods-15-00018-t002:** Ingredients, nutrition facts, and main fatty acid composition (mg FA/100 g) of the three experimental diets used to feed the Canarian laying hens.

Item	SO-d	LO-d	EO-d
Ingredients (%)			
Soybean oil	1.25	---	---
Linseed oil	---	1.10	0.25
*Echium plantagineum* oil	---	---	1.00
Beef tallow	---	0.15	---
Basal mixture ^1^	98.75	98.75	98.75
Calculated nutrition facts			
Metabolizable energy (Kcal/100 g)	273.56	272.84	274.91
Crude carbohydrates (%)	40.93	40.93	40.93
Crude protein (%)	16.48	16.48	16.48
Moisture (%)	12.27	12.27	12.27
Crude ash (%)	12.67	12.67	12.67
Crude fiber (%)	3.6	3.6	3.6
Calcium (%)	4	4	4
Phosphorus (%)	0.39	0.39	0.39
Analyzed nutrition facts			
Crude fat (%)	3.68	3.60	3.83
Main fatty acid composition			
∑SFA	429.45	516.32	461.72
16:0	313.86	355.74	338.34
18:0	85.49	121.89	90.48
∑MUFA	582.28	720.14	679.68
18:1n-9	523.27	633.18	614.38
18:1n-7	35.36	49.34	35.52
∑n-6 PUFA	1085.04	859.00	1100.62
18:2n-6	1083.53	855.04	1025.35
18:3n-6	1.51	3.96	75.27
∑n-3 PUFA	82.04	415.17	379.11
18:3n-3	80.10	412.99	302.33
18:4n-3	1.94	2.18	76.78
n-6/n-3	13.31	2.19	2.90

SO-d, soy oil diet; LO-d, linseed oil diet; EO-d, *Echium* oil diet. FA, fatty acids; SFA, saturated fatty acids; MUFA, monounsaturated fatty acids; PUFA, polyunsaturated fatty acids. Totals include other minor components not shown. ^1^ Basal mixture composed of soybean meal, wheat meal, corn meal, barley meal, phosphate, calcium carbonate, liquid methionine, lysine, salt sodium bicarbonate, axtraXB^®^, fysal^®^, and theonine (Capisa group, Tenerife, Spain).

**Table 3 foods-15-00018-t003:** Productive performance of Canarian laying hens fed diets supplemented with soybean (SO-d), linseed (LO-d), or *Echium* oil (EO-d) for 31 days.

	SO-d	LO-d	EO-d	SEM	*p*-Value
Body weight (kg)	2.39	2.38	2.40	0.058	0.979
Feed intake (g/d)	152.34	165.71	163.33	4.580	0.489
Laying rate (%)	41.07 ^b^	24.43 ^a^	38.10 ^ab^	2.945	0.028
Egg mass (g)	25.56 ^b^	15.98 ^a^	24.58 ^ab^	1.794	0.036
FCR	6.17 ^a^	10.57 ^b^	6.67 ^a^	0.667	0.019

Values are means (*n* = 16). SEM, standard error of the mean. FCR, feed conversion rate. Different letters in the same row indicate significant differences (*p* < 0.05).

**Table 4 foods-15-00018-t004:** Physical and pH characteristics of eggs from Canarian laying hens fed diets supplemented with soybean (SO-d), linseed (LO-d), or *Echium* oil (EO-d) for 31 days.

	SO-d	LO-d	EO-d	SEM	*p*-Value
Egg weight (g)	65.52 ^b^	60.50 ^ab^	59.92 ^a^	1.002	0.034
Egg shape index (%)	73.72	71.42	73.16	0.547	0.222
Egg components (%)					
Yolk	31.25 ^a^	34.41 ^b^	32.36 ^ab^	0.494	0.030
Egg white	56.19	52.26	56.35	0.947	0.151
Eggshell	8.20	8.03	8.03	0.125	0.822
pH yolk	6.42	6.46	6.28	0.036	0.243
pH egg white	9.32	9.31	9.34	0.008	0.346
Yolk color					
L*	61.06	60.17	60.17	0.564	0.802
a*	10.31 ^ab^	9.15 ^a^	12.50 ^b^	0.484	0.019
b*	55.98	56.37	54.08	0.545	0.281
Eggshell color					
L*	84.75	83.79	81.82	0.828	0.337
a*	3.25	3.55	4.65	0.457	0.487
b*	14.91	17.02	17.43	0.771	0.339
Eggshell thickness	0.51 ^b^	0.41 ^a^	0.52 ^b^	0.016	0.001

Values are means (n = 16). SEM, standard error of the mean. Different letters in the same row indicate significant differences (*p* < 0.05).

**Table 5 foods-15-00018-t005:** Total lipid content (% fresh weight) and lipid class composition (% of total lipid) of egg yolk from Canarian laying hens fed diets supplemented with soybean (SO-d), linseed (LO-d), or *Echium* oil (EO-d) for 21 days.

	SO-d	LO-d	EO-d	SEM	*p*-Value
SM	1.11	0.81	0.99	0.100	0.512
PC	13.92 ^b^	15.69 ^b^	10.57 ^a^	0.700	0.002
PI	2.77	2.79	2.76	0.275	0.999
PE	8.10 ^ab^	9.63 ^b^	7.02 ^a^	0.389	0.009
TPL	25.89 ^b^	28.92 ^b^	21.34 ^a^	1.258	0.031
MAG	1.51	1.97	2.10	0.162	0.317
DAG	1.00 ^a^	1.06 ^ab^	1.46 ^b^	0.083	0.036
CHO	18.66	20.07	20.42	0.538	0.397
FFA	1.94	1.34	1.88	0.134	0.136
TAG	46.94	41.34	45.78	1.132	0.094
SE	4.06 ^a^	5.29 ^ab^	7.03 ^b^	0.466	0.018
TNL	74.11 ^a^	71.08 ^a^	78.66 ^b^	1.258	0.031
Total lipid (TL)	31.06	30.34	30.73	0.373	0.762

Values are means (*n* = 5). SEM, standard error of the mean; SM, sphingomyelin; PC, phosphatidylcholine; PI, phosphatidylinositol; PG, phosphatidylglycerol; PE, phosphatidylethanolamine; TPL, total polar lipids; MAG, monoacylglycerols; DAG, diacylglycerols; CHO, cholesterol; FFA, free fatty acids; TAG, triacylglycerols; SE, sterol esters; TNL, total neutral lipids. Different letters in the same row indicate significant differences (*p* < 0.05).

**Table 6 foods-15-00018-t006:** Main fatty acid composition (mg/100 g fresh yolk) and nutritional indexes of egg yolk from Canarian laying hens fed diets supplemented with soybean (SO-d), linseed (LO-d), or *Echium* oil (EO-d) for 21 days.

	SO-d	LO-d	EO-d	SEM	*p*-Value
ΣSFA	7199.14	6624.64	7374.15	140.248	0.061
16:0	5101.56	4600.45	5252.11	120.161	0.055
18:0	1964.26	1905.76	1990.42	36.689	0.662
ΣMUFA	10,136.65	9790.74	10,146.99	231.189	0.799
16:1n-9	217.91	210.79	171.14	11.492	0.210
18:1n-9	8607.16	8532.97	8804.85	203.468	0.871
Σn-6 PUFA	3349.00	2748.21	2771.55	175.385	0.244
18:2n-6	2780.58	2288.82	2226.66	163.785	0.253
18:3n-6	36.62 ^ab^	23.93 ^a^	42.35 ^b^	2.888	0.014
20:2n-6	25.21	29.19	24.78	1.964	0.196
20:3n-6	29.80 ^ab^	27.83 ^a^	43.27 ^b^	3.490	0.036
20:4n-6	374.90	317.86	378.93	13.558	0.115
22:4n-6	33.72	33.90	37.66	1.569	0.546
22:5n-6	68.17 ^b^	26.68 ^a^	17.90 ^a^	6.942	0.001
Σn-3 PUFA	316.00 ^a^	571.20 ^b^	931.40 ^c^	79.700	0.001
18:3n-3	123.27 ^a^	235.37 ^b^	375.80 ^b^	38.391	0.008
18:4n-3	10.94 ^a^	15.72 ^b^	32.88 ^c^	2.607	0.001
20:3n-3	nd	nd	10.14	1.457	0.001
20:5n-3	nd	8.08 ^a^	28.99 ^b^	3.616	0.003
22:5n-3	15.13 ^a^	36.84 ^b^	66.81 ^c^	7.080	0.003
22:6n-3	166.65 ^a^	275.18 ^b^	416.79 ^c^	30.511	0.001
Σn-6 LC-PUFA	531.81	435.45	502.54	17.638	0.060
Σn-3 LC-PUFA	181.78 ^a^	320.11 ^b^	512.59 ^c^	40.135	0.001
Total	22,883.10	21,573.60	23,051.80	317.983	0.109
n-6/n-3	10.88 ^b^	4.75 ^a^	3.12 ^a^	0.946	0.001
EPA + DHA	166.65 ^a^	283.26 ^b^	445.78 ^c^	33.685	0.001
Atherogenic index (AI)	0.39	0.37	0.40	0.010	0.439
Thrombogenic index (TI)	0.93 ^b^	0.82 ^ab^	0.79 ^a^	0.024	0.039
hH	2.34	2.54	2.32	0.068	0.371

Values are means (n = 5). SEM, standard error of the mean. nd, not detected. SFA, saturated fatty acids; MUFA, monounsaturated fatty acids; PUFA, polyunsaturated fatty acids; LC-PUFA, long-chain (≥C20) polyunsaturated fatty acids. Totals include other minor components not shown. Different letters in the same row indicate significant differences (*p* < 0.05).

**Table 7 foods-15-00018-t007:** Sensory analysis of yolk eggs from Canarian laying hens fed diets supplemented with soybean (SO-d), linseed (LO-d), or *Echium* oil (EO-d) for 31 days.

	SO-d	LO-d	EO-d	SEM	*p*-Value
Intensity					
Odor	7.20	6.00	6.00	0.341	0.260
Flavor	6.90	6.10	7.10	0.250	0.230
Aftertaste	6.30	6.50	7.00	0.212	0.396
Yolk color	6.10	5.70	6.50	0.222	0.349
Acceptance					
Odor	7.00	5.70	5.60	0.363	0.219
Flavor	7.20	6.00	6.60	0.331	0.346
Aftertaste	7.10	6.20	6.30	0.317	0.462
Yolk color	6.70	6.10	6.50	0.331	0.766
Texture	6.80	6.20	6.70	0.252	0.598
Overall	7.20	6.00	7.00	0.299	0.220

Values are means (*n* = 10). SEM, standard error of the mean. Each panelist evaluated the attributes with a 9-point intensity scale and then rated the acceptance using a 9-point hedonic scale.

## Data Availability

The original contributions presented in this study are included in the article. Further inquiries can be directed to the corresponding author.

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
