# Peer review of "Enriching Egg Quality of Laying Hens from the Canary Islands by Feeding with *Echium* Oil"

_foods, 2025, doi:10.3390/foods15010018_

Round 1
Reviewer 1 Report
Comments and Suggestions for Authors
The present research offers a potentially interesting perspective to valorize the oil obtained from Echium plantagineum, an endemic plant of Canary Islands, to farm hens from the Canary Islands in the perspective to improve the quality of their eggs (enrichment in n-3 FA).
Despite the potential interesting perspective, the experimental design of the study is not adequately robust for the research.
Specifically, a total of 48 hens were allocated to three different pens (n=16 hens each), corresponding to three different dietary treatments. The problem of this experimental design is that there is no true replication of the treatments and the design is flawed (the true replication should be the single pen, while the single hen within pen is a repetition within replicate). The applied treatment is the experimental diet and in this case the treatment was not replicated (only 1 pen/diet).
Therefore, the experimental design is not adequate for the present research and this is a structural problem.
Then some other key information which would be required in roder to allow a full understanding of the research are:
1) How can an hen of the Canary Islands be defined? Do Authors refer to a particular subspecies, breed, hybrid? Do authors refger to a generic hen born ijn the Canary Islands whose genetic background is unknown? If the latter, this represent another issue since hens can have different genetioc charactEristics which would make productive outputs potentialli very diverse among individuals
2) Why Linseed oil was used in combination with Beef tallow? And Why Echium oil was used in combination with Linseed oil? This masks the real experimental effect associated to the two opils rich in n-3.
3) The lipid content and FA profile of linseed and Echium oil as wella as of beef tallow must be provided in order to correctly interpret results.
Another key issue, in this case with results:
1) The laying rate is much lower in LO-d hens than in SO-d one (41 vs 24 %): how can this be justified?
4) FA content of eggs: how is it possible that eggs of the Control group (n-6) has 123.27 mg/100 g fresh yolk of alpha linolenic acid and > 180 mg/100 g yolk of EPA+DHA if the diet had no relevant sources of n-3? Consider that 80 mg/100 product and 100 kcal are enough to market an egg as omega-3 rich...
Reviewer 2 Report
Comments and Suggestions for Authors
The authors evaluated Echium plantaegineum oil as a local omega-3 source for native Canarian hens and showed that it maintained productive performance while enhancing yolk deposition of key n-3 fatty acids and improving the lipid profile. The study highlights the potential of this endemic oilseed as a plant-based alternative to marine oils for egg enrichment. It is a practical study. Here are some concerns.
1. Title What is the breed of the hen used in this study. The expression ‘from the Canary Islands’ may lack academic rigor.
2. Abstract The abstract mentions “sensory traits” but provides no corresponding results; a brief indication of the findings should be included.
3. The conclusions appear somewhat overstated. Claims regarding “biodiversity conservation” and the “circular economy” are valuable but are not supported by data presented in the abstract.
4. While the EO and LO diets were designed to achieve similar n-6/n-3 ratios, the rationale for the specific inclusion levels of each oil is not fully explained. It would strengthen the manuscript to clarify whether the fatty acid compositions of the EO and LO diets are truly comparable, ensuring that the comparison between these treatments is appropriately balanced.
5. Since the study involves a local hen breed and a native plant oil, it would be helpful for readers if photographs of the hens and Echium plantagineum were provided.
6. It is recommended that letters indicating significant differences be presented as superscripts to improve clarity.
7. The Discussion begins abruptly; a brief summary of the main findings before detailing specific results would improve readability.
8. The Conclusions section could be more concise.
Reviewer 3 Report
Comments and Suggestions for Authors
Dear Authors,
In my opinion, the article „ Enriching egg quality of laying hens from the Canary Islands by feeding Echium oil” (Manuscript ID: foods-4021334) submitted for review contains some interesting information. The following corrections are needed:
L: 110-111 Please specify and describe the 'outdoor area' - whether it was covered with vegetation (if so, what kind), whether it was partially shaded, etc., and most importantly, did this impact the hens' nutrition? Are the authors sure that the hens ate exclusively the feed provided, since they had access to the outdoors?
L: 143-144, L: 146, L: 151, and 154 Please specify how many eggs were analyzed in each group. 15 eggs in each group or 15 total?
L: 154 ‘Eggshell thickness’ - Please specify: how many measurements were taken and in which places on the eggshell?
Please avoid repeating tabular data in the manuscript text (tautology) L: 231, 239
L: 232-234 This is not a correct conclusion because, according to Table 2, there are no statistically significant differences in Laying rate and Egg mass between the LO-d and EO-d groups.
L: ‘Major phospholipids, phosphatidylcholine (PC) and phosphatidylethanolamine (PE) were particularly reduced in EO-d eggs, whereas diacylglycerols (DAG) and sterol esters (SE) rose.’ - compared to which group? According to Table 4, in the case of PE, there are no significant differences between the EO-d and SO-d groups, while in the case of DAG and SE, there are no significant differences between the EO-d and LO-d groups.
L: 20 ‘LO supplementation reduced laying rate and egg mass…’ – but compared to SO group. According to Table 2, there are no statistically significant differences in Laying rate and Egg mass between the LO-d and EO-d groups.
L: 21 ‘… whereas EO produced slightly lighter eggs…’ - compared to the SO group, statistically significantly lighter (see Table 3).
Reviewer 4 Report
Comments and Suggestions for Authors
Manuscript ID: foods-4021334
Title: Enriching egg quality of laying hens from the Canary Islands by feeding Echium oil
The manuscript needs some revision, because there are some aspects of the work that should be corrected and improved. Please, review the following recommendations:
- Lines 26-29: Include a short conclusion to the study at the end of the Abstract, instead of the general sentence that you can merge it into the conclusion.
- Line 56: " [8,9,10,11]" Two references are enough in this sentence.
- Line 295: Please check " El-Zenary et al. [19] " or " El-Zenary et al. [20] " is right?
- Lines 103-104: What does this term "physicochemical" mean here" egg physicochemical quality"?
- Although this term "physicochemical" is present in the study's objective, it is not mentioned again in the manuscript.
- Line 125: "Productive and laying performance " Are they two different things or one thing? If they are two things, what is the difference between them in this study?
- In results section Table 2: Why is laying rate low (24.43- 41.07%) in this study?
- Line 223-227: Move this section "2.9 Ethics Statements" in the start of Materials and Methods
- Authors have to focus on discussing their own findings and interpreting these results. Thus, the discussion section needs serious improvements.
- Line 72: Change "lead" to "lead to"
- Line 91: Change "sympthoms" to "symptoms"
- Line 144: Delete "the" before "egg"
- Line 144: Add "a" before "percentage"
- Line 149: Change "Colorimeter" to "The colorimeter"
- Line 170: Change "by [32]" to "by Galindo et al. [32]"
- Line 191: Change " between days 28-31" to " between days 28 and 31"
- Line 250: Change " total of neutral lipids" to " total neutral lipids"
- Line 293: Change " Egg mass, was" to " Egg mass was"
- Line 297: Change " hen day egg production" to " hen-day egg production"
- Line 375: Change " FA composition, richer in EPA" to " FA composition, which is richer in EPA"
- Line 429: Change " eggs while " to " eggs, while "
- Line 429: Change " remain" to " remained"
- Line 434: Add "the" before "sensory quality"
- Line 441: Change " resources boosting" to " resources, boosting"
- Insert the correct format style for journal in the references in the text and references list.
Round 2
Reviewer 1 Report
Comments and Suggestions for Authors
Authors have provided detailed responses to the previous copmments and have made significant improvements in the manuscript.
To me the manuscript is now acceptable for publication.
Congratulations
Best regards